# Fatty Acid-Binding Proteins 4 and 5 Are Involved in the Pathogenesis of Retinal Vascular Diseases in Different Manners

**DOI:** 10.3390/life12040467

**Published:** 2022-03-23

**Authors:** Megumi Higashide, Masato Furuhashi, Megumi Watanabe, Kaku Itoh, Soma Suzuki, Araya Umetsu, Yuri Tsugeno, Yosuke Ida, Fumihito Hikage, Hiroshi Ohguro

**Affiliations:** 1Departments of Ophthalmology, School of Medicine, Sapporo Medical University, Chuo-ku, Sapporo 060-8556, Japan; megumi.h@sapmed.ac.jp (M.H.); watanabe@sapmed.ac.jp (M.W.); kakuitoh@gmail.com (K.I.); ophthalsoma@sapmed.ac.jp (S.S.); araya.umetsu@sapmed.ac.jp (A.U.); yuri.tsugeno@gmail.com (Y.T.); funky.sonic@gmail.com (Y.I.); fuhika@gmail.com (F.H.); 2Departments of Cardiovascular, Renal and Metabolic Medicine, Sapporo Medical University, Sapporo 060-8556, Japan; furuhasi@sapmed.ac.jp

**Keywords:** fatty acid-binding protein 5 (FABP5), fatty-binding protein 4 (FABP4), vascular endothelial growth factor A (VEGFA), retinal vascular disease (RVD), proliferative diabetic retinopathy (PDR), retinal vein occlusion (RVO)

## Abstract

This study reports on the pathological significance of the vitreous fatty acid-binding protein (Vt-FABP) 4 and 5, and vascular endothelial growth factor A (Vt-VEGFA) in patients with retinal vascular diseases (RVDs) including proliferative diabetic retinopathy (PDR) and retinal vein occlusion (RVO). Subjects with PDR (*n* = 20), RVO (*n* = 10), and controls (epiretinal membrane, *n* = 18) who had undergone vitrectomies were enrolled in this study. The levels of Vt-FABP4, Vt-FABP5, and Vt-VEGFA were evaluated by enzyme-linked immunosorbent assays (ELISA). Retinal circulation levels were measured by a laser-speckle flow analyzer (LSFA) and other relevant data were collected. The Vt-FABP5 levels were significantly (*p* < 0.05) elevated in patients with RVDs compared to control patients. This elevation was more evident in patients with RVO than with PDR. Log Vt-FABP5 was significantly correlated negatively or positively with all the LSFA retinal circulation indexes or Log triglycerides (r = 0.31, *p* = 0.031), respectively. However, the elevations in the Vt-FABP4 and Vt-VEGFA levels were more evident in the PDR group (*p* < 0.05) and these factors were correlated positively with Log fasting glucose and negatively with some of the LSFA retinal circulation indexes. Multivariable regression analyses indicated that the LSFA blood flows of the optic disc at baseline was an independent effector with Log Vt-FABP5 other than several possible factors including age, gender, Log triglycerides, Log Vt-FABP4 and Log Vt-VEGFA. These current findings suggest that Vt-FABP5 is involved in the pathogenesis of RVD in a manner that is different from that for Vt-FABP4 and Vt-VEGFA, presumably by regulating retinal circulation.

## 1. Introduction

Fatty acid-binding proteins (FABPs), a family of molecules, are known to be involved in the intracellular lipid transportation to specific cellular compartments to stimulate several lipid-associated responses such as several signaling, trafficking and membrane synthesis, lipid-mediated transcriptional regulation, and lipid droplets synthesis [1,2,3]. It has been shown that, among these family, FABP4 and FABP5 expressed in macrophages in addition to adipocytes play pivotal roles within the pathogenesis of diabetes mellitus (DM), as well as atherosclerosis [4,5,6,7,8,9], and significant increase of the serum levels of FABP4 and 5 were detected in patients with several cardiovascular and metabolic diseases [10,11,12,13,14,15,16,17,18,19]. Since several of these diseases, including DM and hypertension (HT) are also recognized as being significant risk factors for several retinal vascular diseases (RVDs) including proliferative diabetic retinopathy (PDR), retinal vein occlusion (RVO), and others. It is therefore possible that these FABPs may rationally be involved in these RVDs. In fact, in our recent studies, we reported significant elevations of the vitreous levels of FABP4 (Vt-FABP4) and vascular endothelial growth factor A (VEGFA) (Vt-VEGFA) in patients with RVD; PDR (*p* < 0.001) [20] and RVO (*p* < 0.05) [21], in comparison with those with non-RVD. More interestingly, significantly positive correlations (r = 0.72, *p* < 0.001) in PDR [20] and (r = 0.36, *p* = 0.045) [21] in RVO were identified between both factors. However, several correlation analyses indicated that both factors were evidently regulated differently from in both PDR and RVO, suggesting that FABP4 may play a pivotal role in the etiology of RVDs. Alternatively, these observations prompted us to speculate that FABP5 may also play important roles in the pathophysiology of RVD, similar to FABP4.

However, as of this writing, FABP5 has only been detected within the lens [22] and in tears of ocular tissues [23]. Therefore, in the current study, to elucidate the relationship between FABP5 and RVD, we collected surgical vitreous samples from patients with RVD (PDR or RVO) and non-RVD controls (ERM; epiretinal membranes) patients and determined the FABP5 concentrations in these samples, and correlated the data with systemic factors and retinal circulation indexes that were determined by laser-speckle flow analyzer (LSFA). In addition, the resulting data were compared with analogous data for FABP4 and VEGFA.

## 2. Materials and Methods

The study was conducted according to the guidelines of the Declaration of Helsinki, and approved by the Institutional Review Board of Sapporo Medical University (protocol code 282-76 and 1 October 2017). Informed consent was obtained from all subjects involved in the study.

### 2.1. Patients

Three-port vitrectomy (25 G or 27 G) treated 48 patients (*n* = 48 eyes, mean age 66 ± 10 years; 19 males and 29 females) for RVD; PDR (*n* = 20, mean age 62 ± 9 years; 10 male and 10 female) and RVO (mean age 69 ± 15 years; 3 male and 8 female, BRVO; 7 eyes and CRVO 3 eyes), and 18 non-RVD control patients with epiretinal membrane (ERM, mean age 68 ± 9 years; 7 males and 11 females) were enrolled during 2018 through 2019. For a suitable diagnosis and surgical indication, all patients underwent complete ophthalmologic examinations prior to surgery, and vitrectomy combined with cataract surgery under systemic anesthesia was performed as described previously [20,21]. Except slight vitreous hemorrhaging, no serious post-operative complications were recognized and as of this writing, none of the eyes required reoperation. Although, among patients in the RVD group, 20 patients with PDR who had already been enrolled in our recent study [20] were again enrolled in the current study, analyses by enzyme-linked immunosorbent assay (ELISA) described below of vitreous specimens of these PDR groups were again performed together with the same analyses in the case of the newly enrolled RVO and non-RVD control groups.

Data regarding each patient’s bodily conditions by medical check-ups, in addition to blood pressure and blood cells count and blood biochemistry from peripheral venous blood samples were obtained as described previously [20,21].

### 2.2. Biochemical Measurements

Surgically obtained undiluted vitreous samples from 30 RVD (20 PDR, 10 RVO) and 18 non-RVD control (ERM; epiretinal membrane) subjects were immediately kept at −80 °C until use. Those specimens were then subjected to an analysis by commercially available ELISA kits for FABP 4, FABP 5, or human VEGFA. Determination of the vitreous levels of FABP4 (Vt-FABP4, ng/mg protein), FABP5 (Vt-FABP5, ng/mg protein), or VEGFA (Vt-VEGFA, pg/mg protein) were normalized by the vitreous protein concentrations as described previously [20,21]. Several blood chemistry indices were also performed as described previously [20,21].

### 2.3. Other Analytical Methods

A laser speckle flow analyzer (LSFA) was obtained as described in previous reports [24,25,26]. Briefly, using the mean blur rate (MBR), an index of retinal circulation at a specific site among LSFA images, we focused on the blood circulation levels at the optic disc (OD) of the following four categories; (1) M(A); all areas of the OD, (2) M(V); vascular area of the OD, (3) M(T); tissue area of the OD, and 4) M(V)-M(T). 

Numeric data are expressed as the mean ± SD for normal distributions or medians (interquartile ranges) for skewed variables, and all statistical analyses were processed by JMP 14.3.0 for Macintosh (SAS Institute, Cary, NC, USA) as described previously [20,21].

## 3. Results

As background information of patients with RVD including PDR (*n* = 20), RVO (*n* = 10, BRVO; *n* = 7, CRVO; *n* = 3), and non-RVD controls (epiretinal membrane; *n* = 18), sex, age, body mass index, systemic and diastolic blood pressure, blood chemistry values including total cholesterol, triglycerides, fasting glucose, Hb A1c, BUN, Cr, eGFR, uric acid, AST, ALT, γGTP, and hsCRP were listed in Table 1. Among the three patient groups, (1) fasting glucose and HbA1C values in the PDR group were significantly higher than that for the other groups, (2) BUN in RVO was significantly lower than PDR, and (3) AST in PDR was significantly lower than that for the other groups.

The levels of vitreous FABP5 (Vt-FABP5) were significantly higher in the RVD group than the non-RVD controls (*p* < 0.05), especially in the RVO group (Table 1 and Figure 1A). However, in contrast, Vt-FABP4 and Vt-VEGFA levels also markedly increased in the PDR group but not in the RVO group (Table 1 and Figure 1B,C). Furthermore, the levels of the Vt-FABP5 were not affected by gender (Appendix A), the presence of DM (Appendix A) or HT (Appendix A), whereas gender did not affect Vt-FABP4 or Vt-VEGFA, but both Vt-FABP4 and Vt-VEGFA were significantly influenced in the presence of DM [20]. Since FABPs and VEGFA significantly affect local blood circulation as above, LSFA retinal circulation levels were a quite interesting issue for estimating the clinicopathological aspects of these factors in the present subjects.

Table 2 demonstrated that Log Vt-FABP5 was negatively correlated with all of the LSFA retinal circulation indexes of the ONH at post-operative 1-week; M(A) (r = −0.57 *p* < 0.001), M(V) (r = −0.616 *p* < 0.001), M(T) (r = −0.365 *p* = 0.026) and M(V)-M(T) (r = −0.623 *p* < 0.001, Figure 2B). While, in contrast, Log FABP4 and Log VEGFA were also negatively correlated with only some of the LSFA indexes. Therefore, these data suggest that Vt-FABP5 may play a pathological role within RVD in different manners, as compared to Vt-FABP4 and Vt-VEGFA, that is, Vt-FABP5 or Vt-FABP4 and Vt-VEGFA are more evidently correlated with RVO or PDR pathogenesis, respectively.

To study this issue further, correlation analyses between the levels of each of the three factors and various clinical indices were examined and the results revealed; (1) Log Vt-FABP5 positively correlated with Log triglycerides (r = 0.311, *p* = 0.031), Vt-FABP4 (r = 0.38, *p* = 0.008) or Vt-VEGFA (r = 0.35, *p* = 0.015), (2) Log Vt-FABP4 positively correlated with Log Creatinine (r = 0.34, *p* = 0.019), Log fasting glucose (r = 0.33, *p* = 0.023), Vt-FABP5 (r = 0.38, *p* = 0.008) or Vt-VEGFA (r = 0.68, *p* < 0.001), or negatively correlated with Log AST (r = −0.31, *p* = 0.032), and (3) Log Vt-VEGFA positively correlated with Log fasting glucose (r = 0.39, *p* = 0.007), HbA1C (r = 0.44, *p* = 0.002), Vt-FABP5 (r = 0.35, *p* = 0.015) or Vt-FABP4 (r = 0.68, *p* < 0.001), or negatively correlated with Log AST (r = −0.43, *p* = 0.003) (Table 3 and Figure 2). Additional multivariable regression analysis toward Log Vt-FABP5 by following three models: (1) age, gender, and M(V)-M(T), (2) age, gender, M(V)-M(T), Log Vt-FABP4, and Log Vt-VEGFA, and (3) age, gender, M(V)-M(T), Log Vt-FABP4, Log Vt-VEGFA, and Log Triglycerides as possible determinants demonstrated independent association of M(V)-M(T) with Log Vt-FABP5 (Table 4). Taken together, our current results suggest that FABP5 may independently act as a pivotal regulator for optic disc blood circulation, and have a significant role in the molecular etiology of RVD, especially RVO.

## 4. Discussion

Among the RVDs, PDR, an advanced stage of DR, is recognized as a serious retinal complication of DM among the relatively younger population worldwide [27]. Alternatively, RVO is also a common RVD, and is induced by a thrombosis within the lamina cribrosa manifesting the central (CRVO) or hemi-central (hemi-CRVO), or at an intersection of a branched central retinal artery and vein manifesting branch retinal veins (BRVO) [28,29,30,31,32]. Similar to DR, RVO often associated visual impairment caused by retinal edema and ischemia as well as neovascularization [33]. As the possible molecular pathology of both DR and RVO, VEGF is known to be pivotally involved [34]. Therefore, anti-VEGF agents are rationally administrated intra-vitreously as a supportive therapy prior to vitrectomy for patients with PDR [35] as well as treatment for vision-threatening patients with RVO [36,37,38]. However, despite the fact that such an anti-VEGF therapy induces powerful and beneficial effects, those effects are transient and this mono-therapy usually does not stop the progression of PDR [39,40] and RVO [41,42,43,44]. Therefore, additional therapeutic target molecules, other than VEGF, must be required for RVD. Our recent studies found Vt-FABP4 within patients with RVD and suggested that Vt-FABP4 may be an additional target molecule [20,21]. In addition, in the present study, to our knowledge, this is the first report to demonstrate that another member of the FABP family, namely, FABP5 was also detected in vitreous samples obtained from patients with RVD. Quite interestingly, these levels were significantly higher in PDR patients, especially in RVO patients, as compared to the control patients. Furthermore, correlation analyses and multivariable regression analyses indicated that Vt-FABP5 may differently play within the pathogenesis of RVD by regulating the retinal circulation, with Vt-FABP4 and Vt-VEGFA. Evidently, it was recently revealed that FABP5 upregulates the expression of VEGF in prostate cancer, a key factor that promotes angiogenesis and metastasis [45,46] as well as RVD. Furthermore, a previous report demonstrated that FABP5 is importantly involved in the pathogenesis of the early stages of atherosclerosis [7] which also rationally supports our finding of higher levels of Vt-FABP5 in RVO compared to PDR since atherosclerosis is known to be the most significant risk factor for developing RVO. To support this hypothesis, Vt-FABP5 was more negatively correlated with all LSFA retinal circulation indexes (Table 2 and Table 4). In addition, and interestingly, FABP5 has also been identified within the blood brain barrier (BBB), which is known to be similar to the blood retinal barrier [47], and functions in the transport of docosahexaenoic acid [48,49].

As of this writing, the potential pathological contributions of FABP4, FABP5, and VEGFA within RVDs remain to be elucidated. However, a previous report revealed that VEGFA or basic fibroblast growth factor (bFGF) could induce the expression of FABP4 within endothelial cells, and in turn, the production of FABP4 in endothelial cells facilitates angiogenesis [50]. Furthermore, upon the knockdown of FABP4 within the endothelial cells, the VEGF- and bFGF-induced angiogenesis was significantly inhibited to under baseline conditions [51]. Alternatively, the expression of FABP4 within the endothelial cells is independently inducible with VEGF by oxidative stress [52] and endothelial cell damage [53]. Taken together with the fact that both FABP4 and FABP5 are (1) expressed within the endothelial cells in addition to adipocytes and macrophages [1], and (2) are involved in the physiological and pathophysiological conditions related to several metabolic and cardiovascular diseases [10,17], we rationally speculate that Vt-FABP4 or Vt-FABP5 may solely or, in cooperation with Vt-VEGFA, be involved in the etiology of RVDs following the inflammatory damage of retinal endothelial cells.

As study limitations to this study, following issues require to be investigated. (1) Relatively small patients’ groups (total *n* = 48); however, despite the small number of patients, a moderate correlation between Log triglycerides (r = 0.311, *p* = 0.031), and the LSFA retinal circulation indexes of the ONH were observed. In addition, the origin of Vt-FABP5, the mechanisms responsible for the RVD pathogenesis, and the relationship with FABP4 and VEGFA remain to be elucidated. Thus, additional studies to reveal the molecular mechanisms of FABPs contributing to the RVD and the relationship between Vt-FABP5, Vt-FABP4, Vt-VEGFA, and other unknown factors using larger patients’ groups are being planned. 

## Figures and Tables

**Figure 1 life-12-00467-f001:**
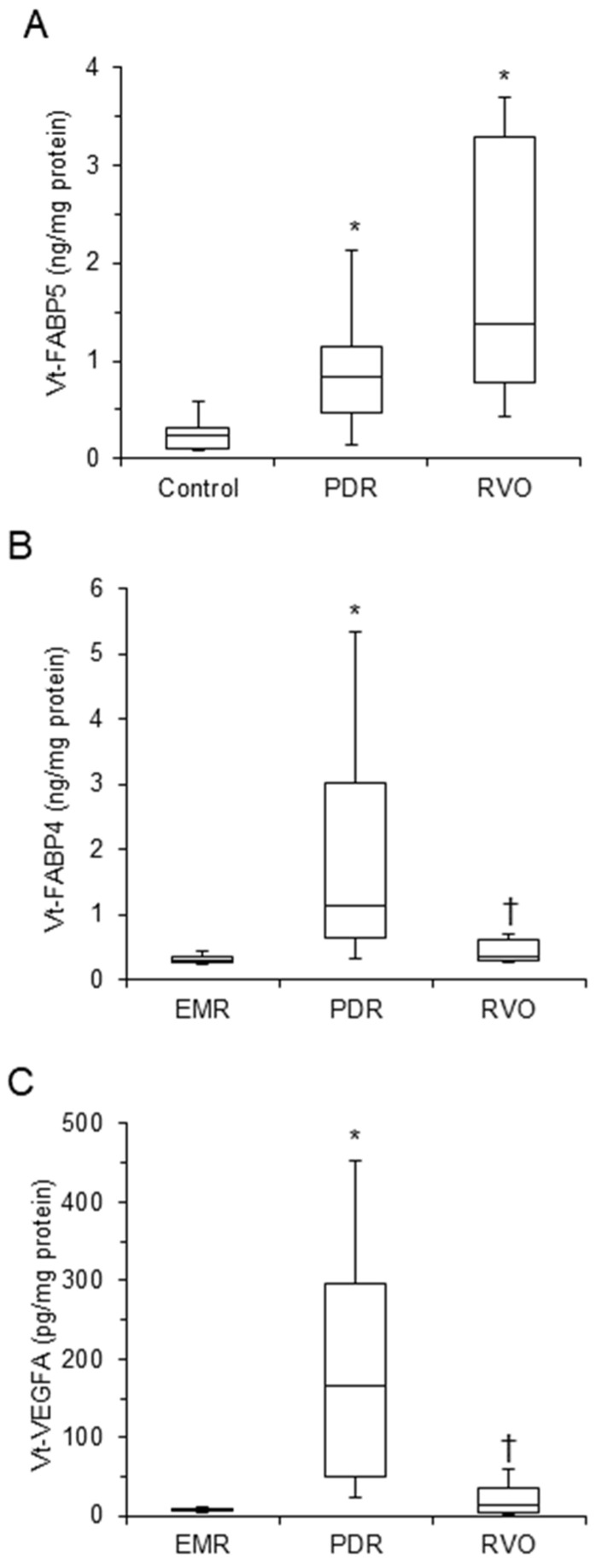
Vitreous concentrations of FABP 5 (Vt-FABP5), FABP4 (Vt-FABP4), and VEGFA (Vt-VEGFA) in patients with non-RVO or RVD. The levels of Vt-FABP5 (ng/mL, **A**), Vt-FABP4 (ng/mL, **B**), or Vt-VEGFA (pg/mL, **C**) from patients with non-RVO control (n= 18, epiretinal membrane) or RVD including PDR (*n* = 20) and RVO (*n* = 10, BRVO; *n* = 7, CRVO; *n* = 3) were measured by ELISA, and were plotted among the three patient groups. * *p* < 0.05 vs. control, † *p* < 0.05 vs. PDR.

**Figure 2 life-12-00467-f002:**
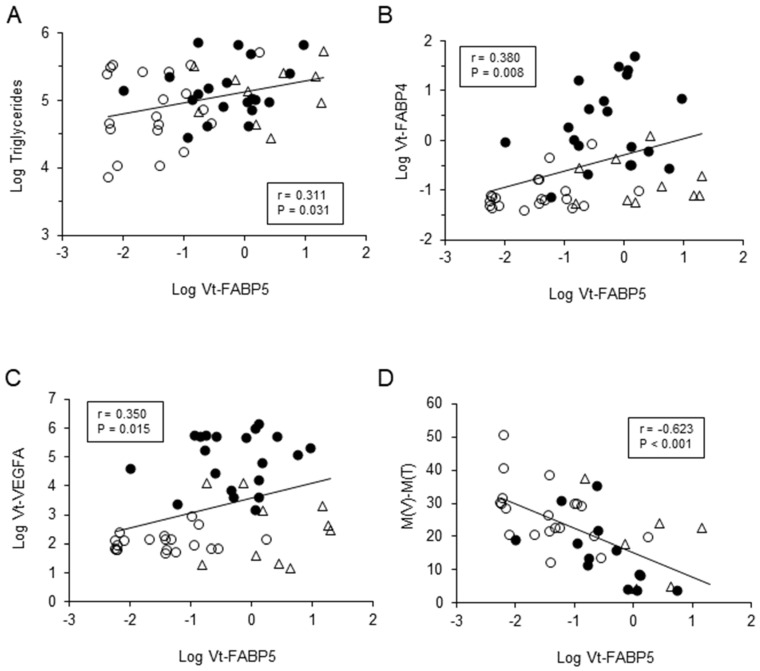
Correlations between Log Vt-FABP5 and Log triglycerides, Log Vt-FABP4, Log Vt-VEGFA or LSFA indexes of the ONH.Log Vt-FABP5 were plotted against Log triglycerides (**A**, r = 0.31, *p* = 0.031), Log Vt-FABP4 (**B**, r = 0.38, *p* = 0.008), Log Vt-VEGFA (**C**, r = 0.35, *p* = 0.015) or M(V)-M(T) at post-surgery week-1 (**D**, r = −0.62, *p* < 0.001) for each subject. Open circles, control subjects; closed circles; subjects with PDR, or open triangles; subjects with RVO.

**Table 1 life-12-00467-t001:** Backgrounds of the patients (*n* = 48).

	Total	Control	PDR	RVO	*p*
*n*	48	18	20	10	
Sex (Male/Female)	19/29	6/12	10/10	3/7	0.453
Age (years)	65 ± 10	68 ± 8	62 ± 9	68 ± 15	0.210
Body mass index	23.8 ± 3.6	23.2 ± 3.4	24.0 ± 4.3	24.6 ± 2.7	0.592
Systolic blood pressure (mmHg)	137 ± 20	137 ± 17	137 ± 25	139 ± 18	0.965
Diastolic blood pressure (mmHg)	78 ± 11	80 ± 10	75 ± 12	82 ± 11	0.201
Biochemical data					
Total cholesterol (mg/dL)	201 ± 38	208 ± 40	195 ± 43	202 ± 18	0.545
Triglycerides (mg/dL)	149 (104–221)	120 (96–222)	157 (136–214)	186 (9120–230)	0.264
Fasting glucose (mg/dL)	126 (105–168)	115 (101–147)	167 (140–184) *	111 (95–122) †	0.011
Hemoglobin A1c (%)	6.4 ± 1.0	6.1 ± 0.9	6.9 ± 1.1 *	6.0 ± 0.5 †	0.006
Blood urea nitrogen (mg/dL)	17 ± 9	15 ± 4	21 ± 12	13 ± 4 †	**0.018**
Creatinine (mg/dL)	0.7 (0.6–0.9)	0.7 (0.6–0.8)	0.8 (0.6–1.0)	0.7 (0.6–0.8)	0.275
eGFR (mL/min/1.73 m^2^)	67.9 ± 24.6	71.0 ± 17.4	63.3 ± 32.5	71.5 ± 16.9	0.562
Uric acid (mg/dL)	5.4 ± 1.3	5.3 ± 1.2	5.6 ± 1.3	4.9 ± 1.4	0.323
AST (IU/L)	21 (16–27)	26 (20–33)	17 (14–22) *	24 (19–33) †	**0.016**
ALT (IU/L)	20 (14–26)	24 (16–29)	16 (11–21)	21 (15–30)	0.071
γGTP (IU/L)	27 (16–51)	26 (15–61)	21 (15–45)	36 (25–57)	0.193
hsCRP (mg/dL)	0.06 (0.04–0.13)	0.06 (0.04–0.12)	0.05 (0.03–0.12)	0.10 (0.05–0.17)	0.488
Vt–FABP4 (ng/mg protein)	0.50 (0.30–0.98)	0.30 (0.26–0.35)	1.14 (0.65–3.03) *	0.36 (0.30–0.61) †	<0.001
Vt–FABP5 (ng/mg protein)	0.24 (0.12–0.48)	0.24 (0.11–0.32)	0.84 (0.47–1.14) *	1.38 (0.77–3.29) *	<0.001
Vt–VEGFA (pg/mg protein)	18.6 (6.2–111.5)	6.8 (5.8––8.4)	166.4 (50.3–295.1) *	12.9 (3.6–35.2) †	<0.001

Data expressed as number, means ± SD or medians (interquartile ranges). * *p* < 0.05 vs. control. † *p* < 0.05 vs. PDR.

**Table 2 life-12-00467-t002:** Correlation analyses for Log Vt-FABP4, Log Vt-FABP5, and Log Vt-VEGFA with retinal circulation (*n* = 37).

	Log Vt-FABP5		Log Vt-FABP4		Log Vt-VEGFA	
	r	*p*	r	*p*	r	*p*
M(A)	−0.57	<0.001	−0.43	0.007	−0.35	0.032
M(V)	−0.62	<0.001	−0.48	0.003	−0.35	0.035
M(T)	−0.37	0.026	−0.14	0.399	−0.13	0.438
M(V)-M(T)	−0.62	<0.001	−0.53	0.001	−0.37	0.023
M(M)	−0.34	0.037	−0.08	0.639	−0.20	0.231

M(A), mean blur rate (MBR) of all of the optic disc (OD); M(M), MBR at the macula; M(T), MBR of tissue area of the OD; M(V), MBR of vascular area of the OD.

**Table 3 life-12-00467-t003:** Correlation analyses for Log Vt-FABP4, Log FABP5, and Log Vt-VEGFA (*n* = 48).

	Log Vt-FABP5		Log Vt-FABP4		Log Vt-VEGFA	
	r	*p*	r	*p*	r	*p*
Age	−0.11	0.449	−0.28	0.051	−0.24	0.096
Body mass index	0.09	0.564	0.13	0.396	−0.04	0.767
Systolic blood pressure	0.13	0.375	0.04	0.807	−0.01	0.958
Diastolic blood pressure	0.01	0.966	−0.24	0.101	−0.18	0.221
Biochemical data						
Log AST	−0.15	0.315	−0.31	0.032	−0.42	0.003
Log ALT	0.04	0.764	−0.17	0.236	−0.24	0.095
Log γGTP	0.14	0.355	−0.16	0.265	−0.14	0.329
BUN	−0.03	0.864	0.26	0.072	0.24	0.102
Log Creatinine	0.11	0.457	0.34	0.019	0.22	0.141
eGFR	−0.08	0.612	−0.24	0.107	−0.07	0.648
Uric acid	0.00	0.978	0.12	0.421	0.24	0.107
Total cholesterol	−0.13	0.388	−0.03	0.825	−0.11	0.451
Log Triglycerides	0.31	0.031	0.20	0.178	0.23	0.116
Log Fasting glucose	0.23	0.111	0.33	0.023	0.39	0.007
Hemoglobin A1c	0.28	0.055	0.26	0.070	0.44	0.002
Log hsCRP	0.23	0.118	0.10	0.516	0.07	0.643
Vitreous humor						
Log Vt-FABP4	0.38	0.008	-	-	0.68	<0.001
Log Vt-VEGFA	0.35	0.015	0.68	<0.001	-	-
Log Vt-FABP5	-	-	0.38	0.008	0.35	0.015

**Table 4 life-12-00467-t004:** Multivariable regression analyses for Log Vt-FABP5.

	Model 1			Model 2			Model 3	
	β	*p*		β	*p*		β	*p*
Age	−0.054	0.687	Age	−0.037	0.793	Age	−0.021	0.878
Sex (Male)	0.146	0.298	Sex (Male)	0.189	0.238	Sex (Male)	0.186	0.242
M(V)-M(T)	−0.583	<0.001	M(V)-M(T)	−0.501	0.006	M(V)-M(T)	−0.488	0.008
			Log Vt-FABP4	0.174	0.432	Log Vt-FABP4	0.176	0.425
			Log Vt-VEGFA	−0.054	0.786	Log Vt-VEGFA	−0.096	0.636
						Log Triglycerides	0.171	0.236
	(R^2^ = 0.412, AIC = 92)			(R^2^ = 0.443, AIC = 97)			(R^2^ = 0.452, AIC = 98)	

## Data Availability

The data that support the findings of this study are available from the corresponding author upon reasonable request.

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
