# Peer review of "Fatty Acid-Binding Proteins 4 and 5 Are Involved in the Pathogenesis of Retinal Vascular Diseases in Different Manners"

_life, 2022, doi:10.3390/life12040467_

Round 1
Reviewer 1 Report
The Authors here proposed a correlation study between the level of certain proteins such as Vt-FABP4, Vt-FABP5 and Vt-VEGFA and retinal vascular diseases (RVDs) in particular proliferative diabetic retinopathy (PDR) and retinal vein occlusion (RVO) patients, suggesting a role in the pathogenesis of those diseases. The paper is well written and the data are clearly presented. However, I would like to suggest some modifications:
-The Vt-FABP5 levels was significantly (P< 0.05) elevated in patients with RVDs compared to control patients. It should be “ levels were”
-Alternatively, these our observation prompt us to speculate that FABP5 may also play important roles in the pathophysiology of RVD, similar to FABP4. It should be “these observations”
- For a suitable diagnosis and surgical indication, all patients underwent complete ophthalmologic examinations in prior to surgery. It should be “examinations prior”
- Among the RVDs, PDR, an advanced stage of DR, a serious retinal complication of DM among the relatively younger population worldwide. This sentence is not clear.
-Therefore, anti-VEGF agents are rationally administrated intra-vitreously. I think administrated is “administered”
- However, despite that such anti-VEGF therapy induce a powerful and beneficial effects. It should be add an s to “induce” and eliminate such
- To support this hypothesis, Vt-FABP5 was more correlated negatively with all LSFA retinal circulation indexes (Tables 2 and 4). It should be better to write “negatively correlated” instead of “correlated negatively”
-However, despite of such small patients’ groups, quite significant correlations between Log triglycerides (r=0.311, P=0.031). This sentence is not scientific correct: I suggest to eliminate “of such” and the correlation is not “quite significant” but moderate because r>0.2 is conventionally defined as a moderate correlation.
- There should be a better discussion of the potential role of the examined proteins (Vt-FABP4, Vt-FABP5 and Vt-VEGFA) in the pathogenetic mechanisms of the retinal vascular diseases analyzed in the discussion section.
Author Response
Dear Editor,
Thank you very much again for the constructive comments concerning our manuscript; " Fatty acid-binding proteins 4 and 5 are involved in the pathogenesis of retinal vascular diseases in different manners“. We carefully examined all of the comments from the Reviewer and have made a series of specific changes in our paper that were suggested by the reviewer. The changes that were made are highlighted and our responses to the comments are listed below;
Reviewer 1
The Authors here proposed a correlation study between the level of certain proteins such as Vt-FABP4, Vt-FABP5 and Vt-VEGFA and retinal vascular diseases (RVDs) in particular proliferative diabetic retinopathy (PDR) and retinal vein occlusion (RVO) patients, suggesting a role in the pathogenesis of those diseases. The paper is well written and the data are clearly presented. However, I would like to suggest some modifications:
- The Vt-FABP5 levels was significantly (P< 0.05) elevated in patients with RVDs compared to control patients. It should be “ levels were”
Answer; Thank you for this comment. As suggested, that was changed to “were”.
- Alternatively, these our observation prompt us to speculate that FABP5 may also play important roles in the pathophysiology of RVD, similar to FABP4. It should be “these observations”
Answer; Thank you for this comment. As suggested, that was changed to “these observations”
- For a suitable diagnosis and surgical indication, all patients underwent complete ophthalmologic examinations in prior to surgery. It should be “examinations prior”
Answer; Thank you for this comment. As suggested, that was changed to “examinations prior”.
- Among the RVDs, PDR, an advanced stage of DR, a serious retinal complication of DM among the relatively younger population worldwide. This sentence is not clear.
Answer; Thank you for this comment. As suggested, that was changed to “Among the RVDs, PDR, an advanced stage of DR, is recognized as a serious retinal complication of DM among the relatively younger population worldwide”.
- Therefore, anti-VEGF agents are rationally administrated intra-vitreously. I think administrated is “administered”
Answer; Thank you for this comment. As suggested, this was changed to “administered”.
- However, despite that such anti-VEGF therapy induce a powerful and beneficial effects. It should be add an s to “induce” and eliminate such
Answer; Thank you for this comment. As suggested, this was changed to “However, despite that anti-VEGF therapy induces a powerful and beneficial effects”
- To support this hypothesis, Vt-FABP5 was more correlated negatively with all LSFA retinal circulation indexes (Tables 2 and 4). It should be better to write “negatively correlated” instead of “correlated negatively”
Answer; Thank you for this comment. As suggested, this was changed to “negatively correlated”.
- However, despite of such small patients’ groups, quite significant correlations between Log triglycerides (r=0.311, P=0.031). This sentence is not scientific correct: I suggest to eliminate “of such” and the correlation is not “quite significant” but moderate because r>0.2 is conventionally defined as a moderate correlation.
Answer; Thank you for this comment. As suggested, this was changed to “However, despite of small patients’ groups, a moderate correlation between Log triglycerides (r=0.311, P=0.031)”
- There should be a better discussion of the potential role of the examined proteins (Vt-FABP4, Vt-FABP5 and Vt-VEGFA) in the pathogenetic mechanisms of the retinal vascular diseases analyzed in the discussion section.
Answer; Thank you for this comment. As suggested, possible mechanisms responsible for these factors within RVDs were included within the 2nd paragraph of Discussion; “As of this writing, the potential pathological contributions of FABP4, FABP5 and VEGFA within RVDs remain to be elucidated. However, a previous report revealed that VEGFA or basic fibroblast growth factor (bFGF) could induce the expression of FABP4 within endothelial cells, and in turn, the production of FABP4 in endothelial cells facilitates angiogenesis [50]. Furthermore, upon the knockdown of FABP4 within the endothelial cells, the VEGF and bFGF induced angiogenesis was significantly inhibited to under baseline conditions [51]. Alternatively, the expression of FABP4 within the endothelial cells is independently inducible with VEGF by oxidative stress [52] and endothelial cell damage [53]. Taken together with the fact that both FABP4 and FABP5 are 1) expressed within the endothelial cells in addition to adipocytes and macrophages [1], and 2) are involved in the physiological and pathophysiological conditions related to several metabolic and cardiovascular diseases [10, 17], we rationally speculate that Vt-FABP4 or Vt-FABP5 may solely or, in cooperation with Vt-VEGFA, be involved in the etiology of RVDs following the inflammatory damage of retinal endothelial cells.”.

Reviewer 2 Report
This paper concerns the pathological significance of the vitreous fatty acid-binding protein (Vt-FABP) 4 and 5, and vascular endothelial growth factor A (Vt-VEGFA) in patients with retinal vascular diseases (RVDs) including proliferative diabetic retinopathy (PDR) and retinal vein occlusion (RVO) is worthy to publish. The paper is well designed, well prepared, the study design and results are well described. However, there are several English grammars mistakes and the whole manuscript should be checked by an English native speaker, i.e.:
- Please change the phrase: "The levels of Vt-FABP4, Vt-FABP5 and Vt-VEGFA were evalusted by enzyme-linked immunosorbent assays (ELISA)." for "The levels of Vt-FABP4, Vt-FABP5, and Vt-VEGFA
were evaluated by enzyme-linked immunosorbent assays (ELISA). (page 1)
- change: "The Vt-FABP5 levels was significantly (P< 0.05) elevated in patients with RVDs compared to control patients." for "The Vt-FABP5 levels were significantly (P< 0.05) elevated in patients with RVDs compared to control patients."
- change the phrase "Since several these diseases, such as DM and hypertension (HT) are also recognized as significant risk factors for several retinal vascular diseases (RVDs)..." for "Since several of these diseases, such as DM and hypertension (HT) are also recognized as significant risk factors for several retinal vascular diseases (RVDs).."
- change "Alternatively, these our observation prompt us to speculate that FABP5 may also play important roles in the pathophysiology of RVD, similar to FABP4." for "Alternatively, these our observations prompt us to speculate that FABP5 may also play important roles in the pathophysiology of RVD, similar to FABP4."
- change "Laser speckle flow analyzer (LSFA) were obtained as described previously [24-26]." for "Laser speckle flow analyzer (LSFA) was obtained as described previously [24-26]."
- change the phrase "Therefore, anti-VEGF agents are rationally administrated intra-vitreously as an supportive therapy before vitrectomy for patients with PDR..." for "Therefore, anti-VEGF agents are rationally administrated intra-vitreously as a supportive therapy before vitrectomy for patients with PDR."
- change the phrase "However, despite that such anti-VEGF therapy induce a powerful and beneficial effects, those effects are transient and these mono-therapy usually could not stop the progression of PDR [39,40] and RVO [41-44]." for "However, despite that such anti-VEGF therapy induce powerful and beneficial effects, those effects are transient and this mono-therapy usually could not stop the progression of PDR [39,40] and RVO [41-44]."
Author Response
Dear Editor,
Thank you very much again for the constructive comments concerning our manuscript; " Fatty acid-binding proteins 4 and 5 are involved in the pathogenesis of retinal vascular diseases in different manners“. We carefully examined all of the comments from the Reviewer and have made a series of specific changes in our paper that were suggested by the reviewer. The changes that were made are highlighted and our responses to the comments are listed below;
Reviewer 2
This paper concerns the pathological significance of the vitreous fatty acid-binding protein (Vt-FABP) 4 and 5, and vascular endothelial growth factor A (Vt-VEGFA) in patients with retinal vascular diseases (RVDs) including proliferative diabetic retinopathy (PDR) and retinal vein occlusion (RVO) is worthy to publish. The paper is well designed, well prepared, the study design and results are well described. However, there are several English grammars mistakes and the whole manuscript should be checked by an English native speaker, i.e.:
Answer; Thank you for this comment. As suggested, the quality of the English used in this manuscript was checked by an English native speaking scientist, Dr. Milton.
- Please change the phrase: "The levels of Vt-FABP4, Vt-FABP5 and Vt-VEGFA were evalusted by enzyme-linked immunosorbent assays (ELISA)." for "The levels of Vt-FABP4, Vt-FABP5, and Vt-VEGFA were evaluated by enzyme-linked immunosorbent assays (ELISA). (page 1)
Answer; Thank you for this comment. As suggested, this was changed to “The levels of Vt-FABP4, Vt-FABP5, and Vt-VEGFA were evaluated by enzyme-linked immunosorbent assays (ELISA)”
- - change: "The Vt-FABP5 levels was significantly (P< 0.05) elevated in patients with RVDs compared to control patients." for "The Vt-FABP5 levels were significantly (P< 0.05) elevated in patients with RVDs compared to control patients."
Answer; Thank you for this comment. As suggested, this was changed to “The Vt-FABP5 levels were significantly (P< 0.05) elevated in patients with RVDs compared to control patients.”
- change the phrase "Since several these diseases, such as DM and hypertension (HT) are also recognized as significant risk factors for several retinal vascular diseases (RVDs)..." for "Since several of these diseases, such as DM and hypertension (HT) are also recognized as significant risk factors for several retinal vascular diseases (RVDs).."
Answer; Thank you for this comment. As suggested, this was changed to “Since several of these diseases, including DM and hypertension (HT) are also recognized as being significant risk factors for several retinal vascular diseases (RVDs)”, additionally based upon English editing by an English native speaking scientist, Dr. Milton.
- change "Alternatively, these our observation prompt us to speculate that FABP5 may also play important roles in the pathophysiology of RVD, similar to FABP4." for "Alternatively, these our observations prompt us to speculate that FABP5 may also play important roles in the pathophysiology of RVD, similar to FABP4."
Answer; Thank you for this comment. As suggested, this was changed to “Alternatively, these our observations prompt us to speculate that FABP5 may also play important roles in the pathophysiology of RVD, similar to FABP4.”
- change "Laser speckle flow analyzer (LSFA) were obtained as described previously [24-26]." for "Laser speckle flow analyzer (LSFA) was obtained as described previously [24-26]."
Answer; Thank you for this comment. As suggested, this was changed to “Laser speckle flow analyzer (LSFA) was obtained as described previously [24-26].”
- change the phrase "Therefore, anti-VEGF agents are rationally administrated intra-vitreously as an supportive therapy before vitrectomy for patients with PDR..." for "Therefore, anti-VEGF agents are rationally administrated intra-vitreously as a supportive therapy before vitrectomy for patients with PDR."
Answer; Thank you for this comment. As suggested, this was changed to “Therefore, anti-VEGF agents are rationally administrated intra-vitreously as a supportive therapy before vitrectomy for patients with PDR.”
- change the phrase "However, despite that such anti-VEGF therapy induce a powerful and beneficial effects, those effects are transient and these mono-therapy usually could not stop the progression of PDR [39,40] and RVO [41-44]." for "However, despite that such anti-VEGF therapy induce powerful and beneficial effects, those effects are transient and this mono-therapy usually could not stop the progression of PDR [39,40] and RVO [41-44]."
Answer; Thank you for this comment. As suggested, this was changed to “However, despite the fact that such an anti-VEGF therapy induces powerful and beneficial effects, those effects are transient and this mono-therapy usually does not stop the progression of PDR [39, 40] and RVO [41-44]” additionally based upon English editing by an English native speaking scientist, Dr. Milton.

This manuscript is a resubmission of an earlier submission. The following is a list of the peer review reports and author responses from that submission.